# AVQACL-MoE: Anchor-Based Mixture-of-Experts for Audio-Visual Question Answering Continual Learning

## Abstract

Audio-Visual Question Answering Continual Learning (AVQACL) aims to enable models to adapt to new tasks while preserving prior knowledge. Existing methods face two primary challenges: Firstly, fine-tuning induces catastrophic forgetting due to cross-task interference in shared backbone parameters, as gradient signals from diverse tasks conflict and overwrite prior knowledge; Secondly, most approaches rely on task labels at inference, which is impractical in open-world settings where task boundaries are fluid and unlabeled. To address these challenges, we introduce AVQACL-MoE, an anchor-based Mixture-of-Experts (MoE) framework that reframes continual learning as incremental expert composition within a frozen pre-trained audio-visual backbone. First, we train corresponding task-specialized experts for different tasks respectively and freeze their parameters after training, structurally reducing the catastrophic forgetting caused by shared backbone parameters. Second, to effectively select the corresponding task-specialized experts, we refine the task signatures into lightweight, modal-specific anchors during training. Then implement task-independent routing based on cosine similarity between the input sample and anchors for inference. On the AVQACL benchmark, AVQACL-MoE achieves state-of-the-art performance, reducing forgetting from 27% to 2% and improving final accuracy by over 30.9%. By shifting the stability-plasticity trade-off from weight adaptation to expert assembly, our approach enables scalable continual learning for open-world AVQA applications. The code is available at https://anonymous.4open.science/r/AVQACL-MoE-C454/.

## 1 Introduction

Continuous Learning (CL) aims to enable models to acquire new knowledge without reducing the knowledge they have already learned, providing theoretical support for the deployment of artificial intelligence in an open real-world environment. Especially for Audio-Visual Question Answering (AVQA) tasks, the real-world AVQA systems encounter an ever-expanding sequence of tasks in unknown order, such as localizing sound sources in cluttered scenes, aligning asynchronous events across modalities, or inferring causal audio-visual relationships. Each task demands distinct cross-modal fusion patterns, making stable and scalable learning especially challenging.

The recently proposed AVQACL benchmark (Wu et al., 2025a) evaluates continual learning in AVQA using deep models, revealing severe catastrophic forgetting under sequential fine-tuning: the best prior method suffers a 24.7% accuracy drop on earlier tasks. However, existing AVQACL methods face several amplified challenges in deep audio-visual architectures. First, sequential fine-tuning causes severe cross-task interference due to shared backbone parameters, as gradient signals from diverse tasks (e.g., spatial localization vs. causal inference) conflict and overwrite prior knowledge Ahn et al. (2021); Pian et al. (2023); Wu et al. (2025a). Second, most studies assume task labels are available at inference, which is unrealistic in open-world settings where task boundaries (Tiwari et al., 2025; Du et al., 2024; Zhu et al., 2024) are fluid and unlabeled (e.g., between counting instruments, locating solos, and inferring temporal order).

To address the challenges mentioned, we propose AVQACL-MoE, which reframes continual learning as the incremental composition of frozen, task-specialized experts within a pre-trained audio-

visual backbone. Firstly, to eliminate cross-task interference from sequential fine-tuning, we train task-specialized experts by replacing intermediate feed-forward network (FFN) layers exclusively for each new task. Then, we freeze the trained task-specialized experts that prevent knowledge overwriting and mitigate catastrophic forgetting via parameter isolation. Secondly, to enable task-independent inference without relying on task labels, we first distill task signatures into lightweight and modality-specific anchors during the training period. Then, we use soft routing based on cosine similarity to select the most relevant task-specialized expert for task inference. Finally, to avoid the rigidity and underutilization of hard-coded Mixture-of-Experts (MoE) designs, our framework supports continual expert accumulation, where the growing pool enables combinatorial knowledge reuse for novel audio-visual queries while maintaining constant inference cost through deterministic top-1 selection.

In summary, by recasting the stability-plasticity dilemma as an expert assembly problem, our AVQACL-MoE facilitates sustainable knowledge expansion: new knowledge is incorporated by adding experts without modifying existing ones, thereby scaling capacity while ensuring prior knowledge retention. The contributions of our method are summarized as follows:

1. We propose a MoE framework specifically designed for the AVQACL task, which enables the performance of task-specialized expert selection and task-independent reasoning based on anchor routing without relying on additional task identifiers.

2. We consider training corresponding task experts for different tasks. Through this structural isolation, our model can prevent knowledge coverage and mitigate catastrophic forgetting while maintaining outstanding performance.

3. Extensive experiments on the AVQACL benchmark show that AVQACL-MoE achieves state-of-the-art performance, with a reduction in average forgetting from 27% to 2% and improving final accuracy by over 30.9%.

## 2 RELATED WORK

### 2.1 AUDIO-VISUAL QUESTION ANSWERING

State-of-the-art AVQA systems learn joint audio-visual representations to answer questions grounded in complex scenes (Yang et al., 2022; Guangyao li, 2022). Early datasets (AVQA (Yang et al., 2022), MUSIC-AVQA (Guangyao li, 2022)) have driven innovations in multimodal attention and cross-modal alignment Li et al. (2023). However, they assume a static training distribution and cannot cope with real-world data streams where new reasoning skills must be acquired on the fly. Recent work addressed biases in these datasets for better generalizability, such as Liu et al. (2024b), which crafted a balanced MUSIC-AVQA for unbiased question-answering, and Ma et al. (2024), which overcame multimodal biases in AVQA, highlighting the need for robust evaluations in dynamic settings. The recently introduced AVQACL benchmark (Wu et al., 2025a) formalized continual learning in AVQA, revealing that naïve sequential fine-tuning leads to catastrophic forgetting ($> 24\%$ accuracy drop). While Question-Aware Gaussian Experts (Kim et al., 2025) improved static AVQA via modality-specific experts, they were not designed for continual adaptation. Similarly, Cheng et al. (2024) proposed an MoE with adapters for audio-visual learning, focusing on unimodal and cross-modal specialization. However, it targeted static scenarios without addressing continual learning issues like forgetting. To the best of our knowledge, this study is the first work addressing AVQA continual learning with a task-agnostic, expert-based architecture that prevents forgetting by construction.

### 2.2 UNIMODAL CONTINUAL LEARNING

Unimodal Continual Learning (UCL) methods mitigated forgetting through parameter regularization (EWC (Kirkpatrick et al., 2017), MAS (Aljundi et al., 2018)), knowledge distillation (Li & Hoiem, 2018; Dhar et al., 2019), or episodic replay (iCaRL (Rebuffi et al., 2017), SS-IL (Ahn et al., 2021)). These techniques excelled on single-modal tasks but struggled with multimodal inputs: distillation losses failed to preserve intricate spatiotemporal relationships across audio and video, while replay incurred prohibitive storage costs for high-resolution videos. Recent surveys (Guo et al., 2025)

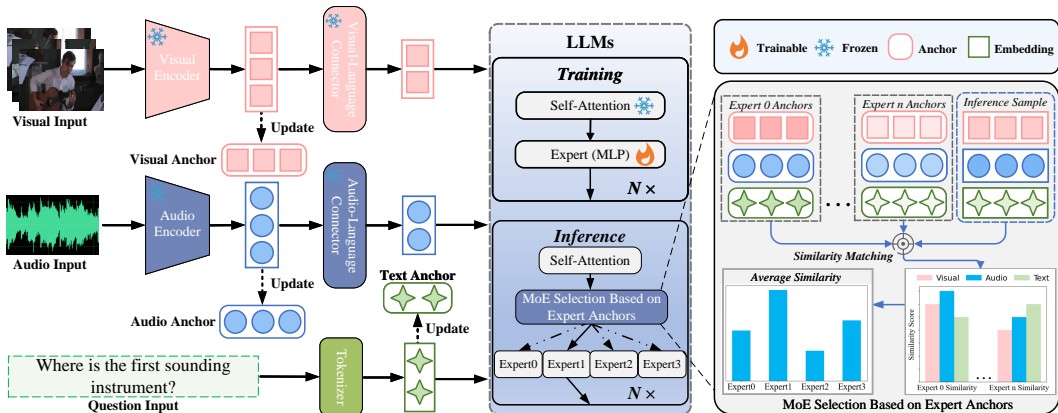

Figure 1: Overview of the AVQACL-MoE Framework. Firstly, we add task-specific expert modules to the LLM structure to isolate the mutual interference between different tasks. We then construct visual, audio, and text anchors to record the tasks' characteristics during the training period. During reasoning, the MoE selection module based on anchor points quickly selects the corresponding task-specialized experts by calculating the similarity between the recorded anchor points and inputs.

confirmed that directly porting UCL methods to multimodal streams was insufficient, motivating our shift from weight adaptation to expert composition.

## 2.3 MULTIMODAL CONTINUAL LEARNING

Multimodal Continual Learning (MCL) research extended continual learning to vision-language and audio-visual domains. vCLIMB (Villa et al., 2022) and PIVOT (Villa et al., 2023) handled incremental video understanding but ignore question-answering. For AVQA-specific continual learning, AVQACL baselines (Wu et al., 2025a) combined question-guided fusion (QCIF), spatiotemporal distillation (TKD-STFC), and semantic consistency constraints (QSCC); yet all were weight-update paradigms exhibiting severe forgetting. Similarly, Nikandrou et al. (2024) enhanced continual learning in visual question answering via modality-aware feature distillation to mitigate forgetting, but relied on distillation that may not capture visual-question interactions fully. Recent MoE continual-learning works (Huai et al., 2025; Yin et al., 2025) required task-ID supervision for routing or pre-assigned experts to single modalities, making them brittle for overlapping tasks and joint audio-visual reasoning. In contrast to learned routers or probabilistic gating (Cheng et al., 2024), which targeted static audio-visual scenarios and require trainable routing networks, our method is designed for continual learning. Their adaptable routers demanded task-specific updates, incurred higher computational costs, and risked overfitting; furthermore. Further, they provided no mechanism to prevent catastrophic forgetting when tasks arrive sequentially. We freeze modality-aware experts, compute lightweight anchors on-the-fly, and route via cosine similarity, eliminating task-ID dependence while enabling fine-grained cross-modal fusion without forgetting.

## 3 METHODOLOGY

To address the above challenges, we propose AVQACL-MOE as shown in Fig. 1, which is a principled continual learning framework for multimodal reasoning that resolves the stability-plasticity dilemma by incrementally composing a sparse mixture of experts. Specifically, we train lightweight task-specialized experts for each task, respectively. Then, during reasoning, the routing mechanism dynamically selects the most relevant expert for each query. With this design, the proposed model monotonically scales its capacity while keeping all previously learned parameters unchanged, mitigating catastrophic forgetting.

Our goal is to obtain a unified model $\mathcal{M}_T$ by learning a series of audio-visual question-answering tasks $\mathcal{T}_{1:T} = \{\mathcal{T}_1, \dots, \mathcal{T}_T\}$. Specifically, we decompose the method into three tightly integrated stages:(1) frozen multimodal alignment via trainable connectors that are pre-trained and then frozen; (2) sparse, task-specialized experts inserted into the frozen transformer; and (3) efficient, anchor-

based expert routing that selects exactly one expert per query at inference. The architectural separation of these stages underpins the theoretical properties discussed in Section A.1.

## 3.1 CROSS-MODAL CONNECTOR TRAINING

Let $d_{\text{hidden}}$ denote the hidden size of the frozen LLM backbone $\mathcal{M}$. The $d_v$ and $d_a$ represent the feature dimensions of the frozen visual and audio encoders $\mathcal{E}_{\text{vis}}$ and $\mathcal{E}_{\text{aud}}$, respectively. We introduce linear connectors that map modality features into the LLM hidden space while preserving pre-trained representations:

$$\mathbf{W}_v \in \mathbb{R}^{d_v \times d_{\text{hidden}}}, \quad \mathbf{W}_a \in \mathbb{R}^{d_a \times d_{\text{hidden}}}. \tag{1}$$

where, we adopt the pre-trained connector $\mathbf{W}_v$ from LLaVA (Liu et al., 2024a) for visual inputs. For audio inputs, we utilize the audio-language connector $\mathbf{W}_a$ trained on audio–text corpora and then freeze it for all subsequent continual learning tasks.

Specifically, for a given video, we first sample $F$ visual frames from it. Then, the visual encoder $\mathcal{E}_{\text{vis}}$ and the audio encoder $\mathcal{E}_{\text{aud}}$ are used to extract the visual and audio features, respectively:

$$\mathbf{V} = \mathcal{E}_{\text{vis}}(\mathbf{x}_{\text{vis}}) \in \mathbb{R}^{(F \cdot N_v) \times d_v}, \quad \mathbf{A} = \mathcal{E}_{\text{aud}}(\mathbf{x}_{\text{aud}}) \in \mathbb{R}^{N_a \times d_a}. \tag{2}$$

where, $N_v$ and $N_a$ represent the number of tokens. Subsequently, the connectors project the obtained features into the LLM hidden space:

$$\tilde{\mathbf{V}} = \mathbf{V}\,\mathbf{W}_v, \qquad \tilde{\mathbf{A}} = \mathbf{A}\,\mathbf{W}_a. \tag{3}$$

## 3.2 TASK-SPECIALIZED EXPERTS TRAINING

When learning new tasks, shared parameter fine-tuning inevitably introduces catastrophic forgetting, which is caused by gradient updates. Based on this, we consider training a task-specialized expert for different tasks to reduce the interference between tasks at the structural level. Specifically, when a new task $\mathcal{T}_i$ arrives, we instantiate a full-capacity expert $\mathcal{E}_i$ by inserting a task-specific MLP module into the LLM with frozen parameters:

$$\mathcal{S}_{\text{expert}} = \{\, l \mid 0 \le l < N,\ l \bmod 2 = 0 \,\}, \tag{4}$$

where $N$ represents the number of layers in the LLM. To ensure the lightweight design, we only insert task-specialized experts in even-numbered layers. For each task-specialized expert, it consists of the lower projection, activation function, and upper projection:

$$\mathcal{E}_{i,l}(\mathbf{h}) = \text{GELU}\left(\mathbf{h}\,\mathbf{W}_{i,l}^{(1)}\right)\mathbf{W}_{i,l}^{(2)}, \tag{5}$$

Finally, for task $i$, we obtained the task-specialized expert weights $\Theta_{\text{train}}^{(i)}$, which contain the knowledge of task $i$:

$$\Theta_{\text{train}}^{(i)} = \{\, \mathbf{W}_{i,l}^{(1)}, \mathbf{W}_{i,l}^{(2)} \,\}_{l \in \mathcal{S}_{\text{expert}}}. \tag{6}$$

During training, we optimize each expert with standard next-token prediction while conditioning on the projected multimodal features:

$$\mathcal{L}_{\text{task}}^{(i)} = -\mathbb{E}_{(\mathbf{x},\mathbf{y}) \sim \mathcal{T}_i} \sum_{t=1}^{|\mathbf{y}|} \log p_{\theta_i}\left(y_t \mid y_{<t}, \tilde{\mathbf{V}}, \tilde{\mathbf{A}}, \mathbf{x}_{\text{text}}\right), \tag{7}$$

where $\theta_i = \Theta_{\text{train}}^{(i)}$. $y_t$ is an prediction at position $t$, and $|\mathbf{y}|$ is the sequence length of the true answer text $y$. Upon convergence, we freeze $\mathcal{E}_i$ and add it to the expert pool; all other parameters remain frozen throughout.

To enable the model to adaptively select the corresponding task-specialized experts, we have refined a compact, parameter-free task-anchor for each expert. Specifically, for mini-batch index $n$, we compute per-modality sample features by averaging valid tokens (padding excluded), and store the

final task anchors for the inference process:

$$\mathbf{f}_{\text{audio}}^{(n)} = \frac{1}{|\mathcal{A}_{\text{valid}}|} \sum_{a \in \mathcal{A}_{\text{valid}}} \left( \mathcal{E}_{\text{aud}}(a) \, \mathbf{W}_a \right), \tag{8}$$

$$\mathbf{f}_{\text{text}}^{(n)} = \frac{1}{|\mathcal{T}_{\text{valid}}|} \sum_{t \in \mathcal{T}_{\text{valid}}} \text{Embed}(t), \tag{9}$$

$$\mathbf{f}_{\text{video}}^{(n)} = \frac{1}{|\mathcal{VI}_{\text{valid}}|} \sum_{vi \in \mathcal{VI}_{\text{valid}}} \left( \mathcal{E}_{\text{vis}}(vi) \, \mathbf{W}_v \right), \tag{10}$$

$$\boldsymbol{\mu}_m^{(n)} = \boldsymbol{\mu}_m^{(n-1)} + \frac{\mathbf{f}_m^{(n)} - \boldsymbol{\mu}_m^{(n-1)}}{n}, \quad m \in \{\text{audio}, \text{text}, \text{video}\}, \tag{11}$$

$$\mathbf{a}_i = \{ \boldsymbol{\mu}_{\text{audio}}, \, \boldsymbol{\mu}_{\text{text}}, \, \boldsymbol{\mu}_{\text{video}} \}. \tag{12}$$

where, $\mathbf{a}_i$ represents the task anchor of expert $i$. Furthermore, due to our unique design, maintaining the anchor only requires $\mathcal{O}(d_{\text{hidden}})$ memory, without extra buffer.

### 3.3 Efficient Expert Routing at Inference

During inference, to effectively select the correct task-specialized expert without using task labels, we adopt a MoE router based on similarity to achieve expert selection. Specifically, we select the appropriate task-specialized experts based on the cosine similarity between the input data and the anchors. First, we calculate the cosine similarity between the current test input and the features of each expert anchor:

$$\rho_{i,m} = \frac{\widehat{\mathbf{s}}_{q,m}^{\top} \widehat{\mathbf{a}}_{i,m}}{\max(\|\widehat{\mathbf{s}}_{q,m}\|_2 \|\widehat{\mathbf{a}}_{i,m}\|_2, \epsilon)}, \quad m \in \{\text{audio}, \text{text}, \text{video}\} \tag{13}$$

where $\widehat{\ }$ denotes $\ell_2$ normalization and $\mathbf{s}_q = \{\mathbf{f}_{q,\text{audio}}, \mathbf{f}_{q,\text{text}}, \mathbf{f}_{q,\text{video}}\}$ denotes the input test data. The small constant $\epsilon$ is used to prevent division by zero.

Subsequently, we aggregate the cosine similarities across the available modalities for each task-specific expert to compute routing weights and select the expert with the highest weight for inference. Let $M_i$ denote the set of modalities (e.g., audio, text, video) present in the inputs for the $i$-th task, as determined by the availability of corresponding features. The routing weight $\rho_i$ for the $i$-th expert is computed as the mean of modality-specific cosine similarities $\rho_{i,m}$ (defined in Eq. equation 13), and the optimal expert is selected as follows:

$$\rho_i = \frac{1}{|M_i|} \sum_{m \in M_i} \rho_{i,m}, \tag{14}$$

$$k^* = \arg\max_i \rho_i, \quad i \in \{1, \dots, T\}, \tag{15}$$

where $T$ is the total number of tasks, and $k^*$ indicates the index of the selected expert. Our method eliminates the need to train routers by calculating similarity, reducing resource consumption while ensuring the accuracy of task-specific expert selection.

## 4 Experiments

### 4.1 Dataset

We benchmark AVQACL-MoE on the AVQACL suite (Wu et al., 2025a), which instantiates two continually-learning splits built upon established AVQA corpora. These splits are engineered to reproduce open-world conditions in which new reasoning skills and audio-visual concepts arrive in an unknown order.

**Split-AVQA** is derived from the original AVQA dataset (Yang et al., 2022). It contains 39,314 real-world 10-second videos paired with 41,008 question-answer instances spanning four question types (where, which, happening, and come from) and 2,135 open-vocabulary answers. The continual-learning protocol divides this corpus into $T = 4$ task stages, where each stage corresponds to one question type.

**Split-MUSIC-AVQA** originates from the MUSIC-AVQA dataset (Guangyao li, 2022) and comprises 7,324 60-second music performance clips with 20,774 QA pairs. Questions fall into $T = 5$ categories (counting, location, comparative, temporal, existential) and share a 40-way answer vocabulary.

## 4.2 EVALUATION METRICS

We adopt the AVQACL benchmark's (Wu et al., 2025a) standardized metrics to quantify both cumulative performance and forgetting.

**Mean Accuracy (MA).** After completing training on task $t$, we record the test accuracy on all tasks seen so far. MA is the average over the $T$ incremental checkpoints:

$$\text{MA} = \frac{1}{T} \sum_{t=1}^{T} a_t.$$

where $a_t$ indicates the accuracy of task $t$ that completing the training on task $t$. MA reports the model's sustained global competence as new question semantics and audio-visual concepts are introduced.

**Average Forgetting (AF).** AF measures knowledge loss on earlier tasks when learning later ones. Let $a_{\gamma,i}$ denote accuracy of task $i$ after training task $\gamma$, and $a_{t,i}$ the accuracy of the same task $i$ after training task $t$ $(t > \gamma)$:

$$\text{AF} = \frac{1}{T-1} \sum_{t=2}^{T} \frac{1}{t-1} \sum_{i=1}^{t-1} \max_{\gamma < t}(a_{\gamma,i} - a_{t,i}).$$

where $T$ represents the total number of tasks, while $t$ denotes the index of the current task. The lower AF indicates superior resistance to catastrophic forgetting.

## 4.3 IMPLEMENTATION DETAILS

All experiments were carried out on 8 A100 GPUs. Following Li et al. (2025), we start from publicly released checkpoints: the CLIP-ViT-L/14-336 vision encoder (Radford et al., 2021), the BEATs_iter3 audio encoder (Chen et al., 2023), and the LLaMa-2-7B-chat language model (Touvron et al., 2023). For every incoming task, we run five epochs of AdamW optimization with $\beta_1 = 0.9$, $\beta_2 = 0.999$, $\epsilon = 10^{-8}$, and weight decay 0. The learning rate warms up linearly from 0 to $4 \times 10^{-5}$ with a warmup ratio of 0.03, then follows a cosine schedule. We adopt a global batch size of 8. We evaluate the accuracy of predicted answers using a streamlined process combining exact matching and semantic evaluation. For non-matching pairs, we employ the Qwen-Flash API to assess semantic equivalence, and prompts are modified from (Wu et al., 2025b).

## 4.4 EXPERIMENTAL RESULTS

**Baseline.** We evaluate our proposed method against several representative and state-of-the-art continual learning approaches, including LwF (Li & Hoiem, 2018), EWC (Kirkpatrick et al., 2017), MAS (Aljundi et al., 2018), iCaRL (Rebuffi et al., 2017), SS-IL (Ahn et al., 2021), AV-CIL (Pian et al., 2023), AVQACL (Wu et al., 2025a), along with recent CL-MoE (Huai et al., 2025) with the same LLaVA backbone (without the audio connector we designed).

**Comparison and Analysis.** Table 1 summarizes CL performance on both splits. Our method achieves state-of-the-art performance on both datasets, effectively mitigating catastrophic forgetting in AVQACL. Regularization-based methods (EWC, LwF) and distillation approaches (iCaRL) bring only marginal relief (MA $\leq 25.28\%$, AF $\geq 30\%$), while replay-heavy SS-IL and AV-CIL reduce forgetting to $\sim 2.7\%$ AF, but still plateau below 33% MA. Because their weight-update paradigm cannot reconcile heterogeneous audio-visual fusion patterns, which confirms the severity of catastrophic forgetting in multimodal reasoning.

The prior best AVQACL baseline tops out at 32.05% MA and 2.47% AF on Split-AVQA, yet remains over 54 points behind our result. Recent MoE-based methods like CL-MoE (Huai et al., 2025)

perform poorly on Split-MUSIC-AVQA (25.58% MA), primarily because they are designed for continual visual question answering without explicit support for the audio modality, which is essential for tasks involving sound-source localization, instrument identification, and temporal audio cues in music performance videos. Our AVQACL-MoE shatters this ceiling without any replay buffer ($M = 0$). On the Split-AVQA dataset, it achieves 86.41% MA while keeping average forgetting at an all-time low of 2.34%. This trend is equally striking on the longer-horizon Split-MUSIC-AVQA: we record 72.02% MA, surpassing the strongest competitor by 38 points, and drive AF down to 1.79%. Critically, the gain is obtained without revisiting past data or task labels, validating that freezing the backbone and assembling task-specialized experts is both more effective and more scalable than continual weight adaptation.

Table 1: Experimental results of various methods on the audio-visual question answering continual learning datasets, Split-AVQA and Split-MUSIC-AVQA; M represents the size of memory bank (replay buffer, is used to storing a small fixed size of exemplar memory set of data from previous answers). MA: higher is better, AF: lower is better.

| Method | Split-AVQA | | | Split-MUSIC-AVQA | | |
|---|---|---|---|---|---|---|
| | M | MA(%) | AF(%) | M | MA(%) | AF(%) |
| LwF (Li & Hoiem, 2018) | 0 | 21.53 | 30.49 | 0 | 30.93 | 30.14 |
| EWC (Kirkpatrick et al., 2017) | 0 | 18.89 | 40.69 | 0 | 28.76 | 32.58 |
| MAS (Aljundi et al., 2018) | 0 | 24.47 | 8.91 | 0 | 29.33 | 31.15 |
| iCaRL (Rebuffi et al., 2017) | 5000 | 25.28 | 33.76 | 700 | 28.46 | 33.16 |
| SS-IL (Ahn et al., 2021) | 5000 | 30.50 | 2.68 | 700 | 32.16 | 28.05 |
| AV-CIL (Pian et al., 2023) | 5000 | 27.49 | 2.91 | 700 | 31.22 | 29.65 |
| AVQACL (Wu et al., 2025a) | 5000 | 32.05 | 2.47 | 700 | 33.64 | 27.08 |
| CL-MoE (Huai et al., 2025) | 0 | 29.76 | 3.27 | 0 | 25.58 | 2.45 |
| Ours | 0 | **86.41** | **2.34** | 0 | **72.02** | **1.79** |

## 4.5 ABLATION STUDY

**Impact of Continual Learning Strategy.** To assess the contribution of our CL approach beyond the intrinsic capability of the visual-audio-language LLM backbone, we conduct an ablation by removing the CL components. In this setting, the LLM is directly fine-tuned sequentially on the AVQACL tasks without task-specific experts or anchor-based routing, thus relying solely on its pre-trained multimodal alignment and reasoning.

Table 2: Ablation study on the impact of our continual learning strategy on Split-AVQA and Split-MUSIC(-AVQA). MA: higher is better, AF: lower is better.

| Method | Split-AVQA | | Split-MUSIC | |
|---|---|---|---|---|
| | MA(%) | AF(%) | MA(%) | AF(%) |
| Ours w/o CL | 74.98 | 7.89 | 43.61 | 41.54 |
| Ours | **86.41** | **2.34** | **72.02** | **1.79** |

Table 3: Task accuracy matrix on Split-MUSIC-AVQA without CL. Each entry shows the accuracy of a task (row) after training up to the corresponding task (column). Severe forgetting is observed, particularly for Task 0 (Counting).

| Task | T0 | T1 | T2 | T3 | T4 |
|---|---|---|---|---|---|
| T0 Counting | 0.6366 | 0.0021 | 0.0756 | 0.0425 | 0.0068 |
| T1 Existential | – | 0.8131 | 0.7804 | 0.5507 | 0.6340 |
| T2 Location | – | – | 0.6346 | 0.1006 | 0.5283 |
| T3 Comparative | – | – | – | 0.5099 | 0.3569 |
| T4 Temporal | – | – | – | – | 0.6546 |

As shown in Table 2, the plain Multimodal LLM (MLLM) with frozen visual-language and audio-language connectors already achieves non-trivial MA, indicating that its inherent knowledge contributes substantially to performance. However, on the Split-MUSIC-AVQA dataset, after instruction tuning on Task 1 (Existential), which mainly requires binary answers, the evaluation on Task 0 (Counting) predominantly yields "yes" or "no" instead of numerical predictions, leading to low MA and high AF and thus revealing severe catastrophic forgetting. To provide a finer-grained view, we report the full task accuracy matrix in Table 3, which shows the forgetting pattern: for example, Task 0 (Counting) accuracy drops from 63.7% when trained alone to nearly 0% after subsequent tasks, while the overall MA declines to 43.61% with an AF of 41.54%. These results highlight that sequential fine-tuning without CL causes drastic cross-task performance collapse, whereas our CL strategy with task-specific experts

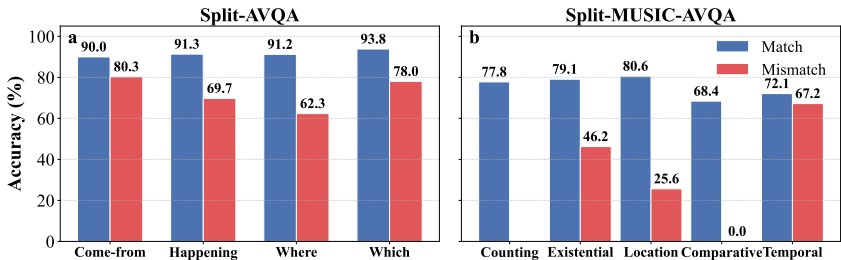

Figure 2: Anchor-guided routing accuracy. Match is the test accuracy that anchors guided to the matching expert. Mismatch is the test accuracy that anchors guided to the other experts.

and anchor-based routing effectively mitigates forgetting and enhances plasticity beyond the raw MLLM.

**Task-Agnostic Anchor Routing: How Faithful Are Our Experts?** We evaluate our anchor-based router on Split-MUSIC-AVQA and Split-AVQA. Fig. 2 reports match and mismatch accuracies, and Fig 3 shows empirical routing distributions. These datasets include overlapping or semantically adjacent tasks, e.g., "Count" versus "Exist" in MUSIC-AVQA, enabling assessment under realistic boundary ambiguity.

On Split-MUSIC-AVQA, anchors route correctly most of the time, with minimal performance drop on mismatched queries, demonstrating graceful degradation. Split-AVQA shows an even higher global match rate, with modest loss on mismatches. This robustness arises from shared cross-task audio-visual patterns, allowing off-target experts to generalize reliably. Anchors computed as running means naturally adapt to gradual task drift, supporting label-free task identification and generalization to few-shot or shifted distributions. Compared to learned routers in existing MoE approaches, our parameter-free anchors offer simplicity, stability, and most importantly, need no buffer replay to train MoE routers in CL scenario.

**The Effect of Multi-modality for Anchor-based Expert Routing.** To evaluate the impact of modality configurations on the effectiveness of our anchor-based routing, we conduct an ablation study using different combinations of modalities, visual (V), visual-audio (VA), and visual-audio-text (VAT), on both datasets. We measure match accuracy (queries routed to the expert trained on the corresponding task). The results are summarized in Fig. 4.

On Split-AVQA, the tri-modal VAT anchor achieves the best overall trade-off, while removing text (VA) substantially reduces performance, and the visual-only (V) anchor performs comparably poorly, indicating that textual signatures play a critical role in refining anchor routing. On Split-MUSIC-AVQA, VAT again clearly dominates, whereas VA and V collapse to very low accuracy. This large gap arises because performance scenes often share both visuals and audio, leaving no

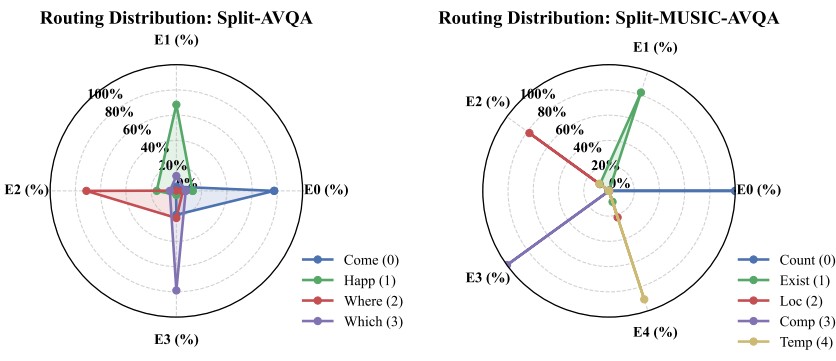

Figure 3: Routing distribution (%) obtained by our anchor-based task-agnostic router on the CL splits. For each task, the figure shows the percentage of test queries directed to every expert.

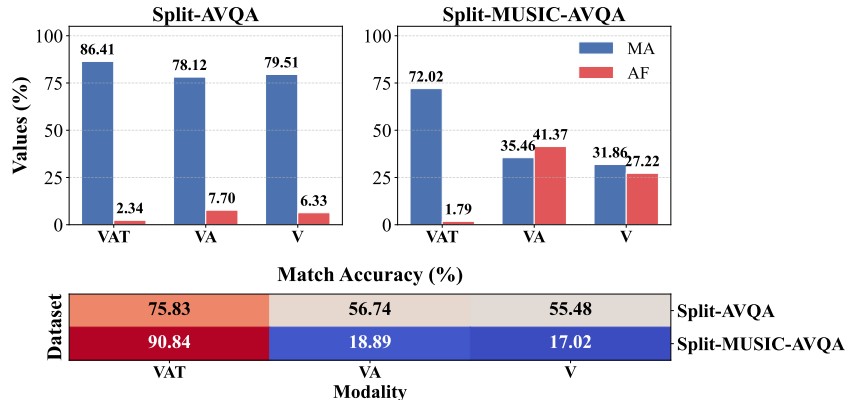

Figure 4: Ablation study on modality configurations for anchor-based routing across Split-AVQA and Split-MUSIC-AVQA datasets. The bar graphs illustrate the performance of modality combinations of V (Visual), A (Audio), and T (Text). The heatmap below correlates match accuracy percentages with three modality configurations in two datasets.

reliable cues for query type; only text disambiguates such cases. These findings demonstrate that multimodal (VAT) anchors are essential for robust task-agnostic routing under diverse continual-learning conditions.

**Impact of Expert Insertion Layer Parity.** We further study how the parity of transformer layers at which experts are inserted affects continual-learning performance. By default, AVQACL-MoE inserts task-specific FFN experts at every *even*-indexed layer (indices $l$ such that $l \bmod 2 = 0$). We conduct an ablation that instead places experts at *odd*-indexed layers, keeping all other settings identical. Table 4 summarizes the results on both continual-learning splits.

Table 4: Insertion-layer parity ablation on Split-AVQA and Split-MUSIC-AVQA. MA: higher is better, AF: lower is better.

| Insertion Layers | Split-AVQA | | Split-MUSIC-AVQA | |
| --- | --- | --- | --- | --- |
| | MA(%) | AF(%) | MA(%) | AF(%) |
| Even (main) | 86.41 | 2.34 | 72.02 | 1.79 |
| Odd | 87.06 | 1.65 | 71.25 | 1.91 |

On Split-AVQA, switching to odd layers slightly lifts MA by 0.65% and further suppresses AF by 0.69%. Conversely, on Split-MUSIC-AVQA, odd layers incur a modest drop of 0.77% in MA and a 0.12% rise in AF. These minor fluctuations indicate that the framework is largely insensitive to the exact layer parity once the backbone is frozen and parameter isolation is enforced.

## 5 CONCLUSION AND LIMITATION

We presented **AVQACL-MoE**, an anchor-based mixture-of-experts framework for audio-visual question answering continual learning. By freezing the multimodal backbone and composing task-specialized experts, our method structurally prevents catastrophic forgetting while scaling to new tasks. The anchor-based routing enables task-agnostic inference without relying on task identifiers, and experiments on the AVQACL benchmark show significant gains in mean accuracy with near-zero forgetting.

A potential limitation is the linear growth of expert pools with task count. However, in practice, AVQA systems typically handle a limited number of core task types. For extended scenarios, similar tasks can be dynamically merged via anchor-driven clustering to cap memory overhead. Future work will explore dynamic parameter-sharing MoE variants.

## ETHICS STATEMENT

Our AVQACL-MoE framework advances continual learning for audio-visual question answering by mitigating catastrophic forgetting and enabling task-agnostic inference. While the approach demonstrates strong performance, it also inherits general risks common to generative models: outputs may occasionally be incorrect or misleading, which could be problematic in safety-critical or decision-making scenarios. In addition, dataset biases (e.g., overrepresentation of certain instruments or scenes) may lead to unfair or skewed predictions. We strongly encourage researchers and practitioners to validate outputs carefully, especially in sensitive domains such as surveillance, education, or assistive technologies. The intended purpose of this work is methodological advancement and academic study, not deployment in high-stakes environments, and we advocate for responsible and transparent use that ensures fairness, privacy protection, and compliance with ethical and legal standards.

## REPRODUCIBILITY STATEMENT

To ensure reproducibility of our findings, we provide detailed descriptions of the model architecture, training configurations, and evaluation protocols in Sections 3 and 4 of the paper, with implementation details in Section 4.3 and theoretical proofs in Appendix A.1. Ablation studies and case analyses (Section 4.5 and Appendix A.2) further support robustness. All source code, training scripts, full hyperparameter settings, and pretrained weights are available at the following anonymous repository: `https://anonymous.4open.science/r/AVQACL-MoE-C454/` to facilitate full replication of our results by other researchers.

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

# A  APPENDIX

## A.1  THEORETICAL PROPERTIES

We now analyze the theoretical guarantees of our AVQACL-MoE framework. These properties stem from the parameter isolation across experts and the anchor-based routing mechanism, which together enable robust continual learning without catastrophic forgetting.

**Setup.** Let $\Psi_i = \{\theta \mid \partial \mathcal{L}_{\text{task}}^{(i)} / \partial \theta \neq \mathbf{0}\}$ denote the parameter influence set for task $\mathcal{T}_i$, representing the subset of model parameters updated during the training of task $\mathcal{T}_i$. Our design enforces parameter isolation, ensuring $\Psi_i \cap \Psi_j = \emptyset$ for all $i \neq j$, and inference routes a query to a single expert via cosine similarity to anchors, as defined in Eq. equation 14.

**Zero Catastrophic Forgetting.**

**Theorem 1.** *Under parameter isolation, the expert $\mathcal{E}_i$ remains unchanged after learning any subsequent task $\mathcal{T}_j$ where $j > i$.*

*Proof.* Since $\Psi_i \cap \Psi_j = \emptyset$ for $i \neq j$, the parameters of $\mathcal{E}_i$ (denoted $\Theta_{\text{train}}^{(i)}$) are not in the gradient path of $\mathcal{L}_{\text{task}}^{(j)}$ and thus receive no updates during the optimization of task $\mathcal{T}_j$. Consequently, the functional mapping of $\mathcal{E}_i$—defined by its forward pass in Eq. equation 5—remains invariant, preserving the performance on task $\mathcal{T}_i$ indefinitely. $\square$

**Routing Stability.** Let $\rho_i(q)$ denote the aggregated cosine similarity for query $q$ with respect to the $i$-th expert, and define the margin $\Delta_i(q) = \min_{j \neq i}(\rho_i(q) - \rho_j(q))$. If $\Delta_i(q) > 0$ for all evaluation queries of task $\mathcal{T}_i$, then the addition of further tasks and their corresponding anchors does not alter the routed index $\arg\max_j \rho_j(q)$ nor the end-to-end prediction. Empirical validation in Section 4.5 confirms positive margins in practice.

**Complexity.** With $|\mathcal{E}_i|$ parameters per expert and $|\mathbf{a}_i| = d_{\text{sig}}$ floats for each anchor, the memory complexity is $\mathcal{M}(T) = \mathcal{M}_{\text{base}} + \sum_{i=2}^{T}(|\mathcal{E}_i|) + \sum_{i=1}^{T}(|\mathbf{a}_i|)$, and the computational complexity is $\mathcal{C}(T) = \underbrace{O(T\,d_{\text{sig}})}_{\text{routing}} + \underbrace{O(|\mathcal{E}|)}_{\text{single forward}}$. The use of hard top-1 selection maintains a forward pass cost independent of $T$, while routing scales linearly with the number of tasks $T$.

**Knowledge Accumulation.** The sets of experts and anchors grow additively: $\mathcal{P}^{(T)} = \mathcal{P}^{(T-1)} \cup \{\mathcal{E}_T\}$ and $\mathcal{A}^{(T)} = \mathcal{A}^{(T-1)} \cup \{\mathbf{a}_T\}$. This design ensures a monotonic increase in model capacity while preserving the functionality of previously learned tasks by construction.

## A.2  CASE STUDY

We further illustrate the behavior of our anchor-based routing through three inference examples without access to task labels (Fig. 5).

(a) *"Where is the loudest instrument?"* — correctly routed to the **Location** expert, producing the answer *left*.

(b) *"Is there a voiceover?"* — incorrectly routed to the **Counting** expert, but predicting the right answer *yes*.

(c) *"What happened in the video?"* — correctly routed to the **Happening** expert, with the answer *Volcanic explosion*.

These cases highlight two key properties of our framework: (i) anchors enable task-agnostic routing, allowing the model to operate without explicit task identifiers; and (ii) even in cases of routing mismatch, the model can still generate semantically plausible outputs by leveraging shared cross-task knowledge. This robustness demonstrates the practicality of our approach in open-world AVQA scenarios where task boundaries are inherently ambiguous.

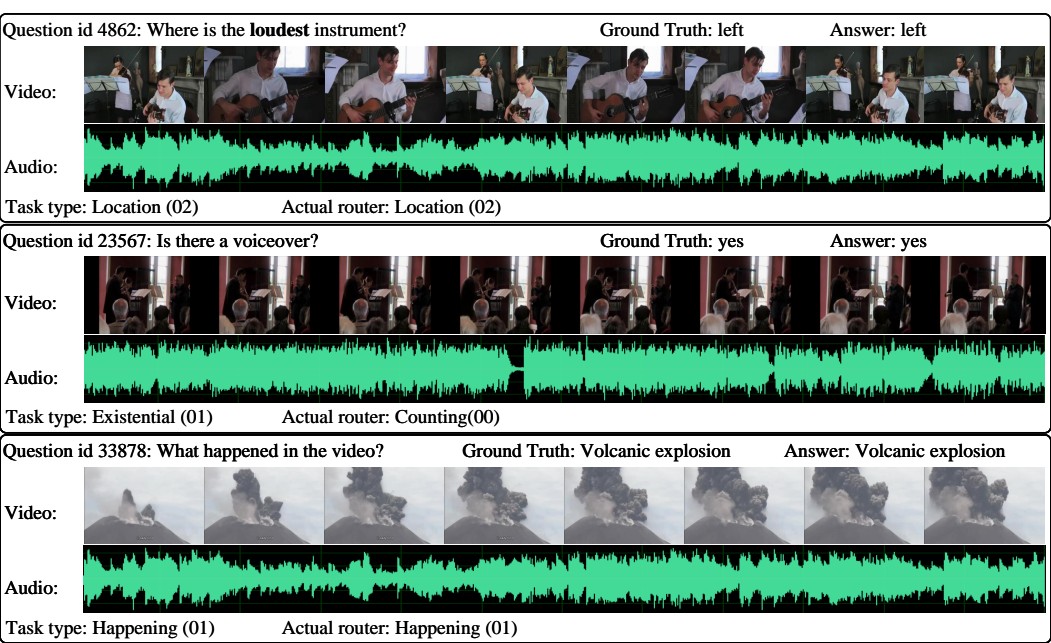

Figure 5: Case Study.

