# OpenReview forum: "AVQACL-MOE: Anchor-Based Mixture-of-Experts for Audio-Visual Question Answering Continual Learning"
_ICLR.cc/2026/Conference — ICLR 2026 Conference Withdrawn Submission_

### Official Review · Reviewer_B1Xt · 2025-10-25

**Soundness:** 3
**Presentation:** 2
**Contribution:** 1
**Rating:** 2
**Confidence:** 4

**Summary:**

In this paper, the authors address the problem of catastrophic forgetting in audio-visual question answering (AVQA) tasks. They propose a continual learning method based on parameter isolation, namely AVQACL-MoE, which leverages a Mixture-of-Experts (MoE) framework to allocate a new expert for each task, avoiding knowledge interference. Furthermore, the method introduces a task anchor mechanism to route relevant experts, eliminating the reliance on task labels during inference.

**Strengths:**

It applies MoE based continual learning to address catastrophic forgetting in AVQA tasks.

**Weaknesses:**

1. While the paper is technically sound, it lacks novelty. The work just uses Mixture-of-Experts (MoE) for parameter isolation based continual learning, and applies them to the AVQA task, which is quite conventional and does not introduce new theoretical insights or methodology innovation.
2. Theoretical and empirical analyses should be provided to justify the superiority of the untrainable allocation strategy.
3. Using each expert to isolate knowledge learning will make the number of experts grow larger when many tasks arrive sequentially in reality. In addition, the method is limited for complex reasoning tasks that need compositional knowledge for joint leaning instead of a single piece of knowledge for a specific task.

**Questions:**

1. What are the main contributions and novelty compared with previous studies?
2. The meaning of Eq. (11) needs to be clearly interpreted in the paper.
3. The theoretical and empirical analysis should be provided to prove the superiority of the untrainable allocation strategy.
4. The performance on compositional question answering needs to be investigated.

---

### Official Review · Reviewer_CD92 · 2025-10-28

**Soundness:** 2
**Presentation:** 3
**Contribution:** 1
**Rating:** 2
**Confidence:** 5

**Summary:**

This paper introduces AVQACL-MoE, a Mixture-of-Experts (MoE) framework for audio-visual question answering continual learning (AVQACL). The authors argue that prior AVQACL methods suffer from catastrophic forgetting and task-ID dependency. Their approach freezes a pre-trained multimodal backbone (visual, audio, and language encoders) and introduces task-specialized experts inserted into the LLM’s feed-forward layers. Each expert is associated with an anchor, a lightweight modality-specific signature (audio, visual, and text) used for task-agnostic routing during inference. The model selects the most relevant expert via cosine similarity between input features and stored anchors. Experiments on the AVQACL benchmark (Split-AVQA and Split-MUSIC-AVQA) claim large improvements—mean accuracy rising from 32% to 86% and forgetting reduced to 2%. Ablation studies explore the impact of anchor modalities and layer insertion parity.

**Strengths:**

- The paper clearly identifies the key challenges in continual learning for audio-visual question answering and motivates the problem well.

- The proposed framework is organized in a clear, modular way (connectors, experts, and anchors), and the accompanying figures help make the overall workflow easy to understand.

- Experiments are conducted on both Split-AVQA and Split-MUSIC-AVQA, with additional ablations on modality configurations and expert-insertion layers, showing reasonable experimental coverage.

- The paper is generally well written and easy to follow.

**Weaknesses:**

- The comparison with baselines is not fair. The proposed method uses a completely different backbone (e.g., LLaVA + LLaMA-2 + CLIP + BEATs), while the reported baseline results are directly taken from the AVQACL paper, which used much smaller models. This makes the huge performance gain meaningless and severely undermines the validity of the experimental claims.

- The paper does not specify the dataset used to pretrain the audio connector. Without this information, it’s unclear whether there is any data leakage between the pretraining corpus and the continual learning tasks, which raises concerns about reproducibility and fairness.

- The core technical contribution—“task-specific experts”—is not essentially different from existing adapter-based continual learning methods for multimodal large language models [1, 2]. The only change is placing the adapters in the feed-forward layers instead of the projection or LoRA layers within self-attention, which makes the technical novelty quite limited and incremental.


[1] Continual Instruction Tuning for Large Multimodal Models. arXiv:2311.16206.

[2] CL-MoE: Enhancing Multimodal Large Language Model with Dual Momentum Mixture-of-Experts for Continual Visual Question Answering. In CVPR 2025.

**Questions:**

See Weaknesses.

---

### Official Review · Reviewer_Qa3M · 2025-10-30

**Soundness:** 2
**Presentation:** 3
**Contribution:** 2
**Rating:** 4
**Confidence:** 3

**Summary:**

This paper proposes AVQACL-MoE, an anchor-based Mixture-of-Experts framework for Audio-Visual Question Answering Continual Learning. The method addresses two central challenges: catastrophic forgetting and the reliance on task labels at inference. The approach freezes a pretrained multimodal backbone and incrementally adds task-specialized experts, each trained independently. A novel anchor-based routing mechanism performs task-agnostic expert selection via cosine similarity between input samples and modality-specific anchors. Experiments on Split-AVQA and Split-MUSIC-AVQA show substantial gains, reducing forgetting from 27% to 2% and improving final accuracy by more than 30%.

**Strengths:**

1. Experiments:
The experimental performance is outstanding — the proposed method significantly outperforms existing baselines on two benchmark datasets without using any replay buffer.
The experiments are comprehensive, including extensive ablation studies, theoretical analyses, and case studies.
The work is reproducible, with complete open-source code provided.
2. Method:
The combination of a frozen backbone and lightweight expert modules effectively mitigates catastrophic forgetting caused by parameter interference.
The innovative anchor-based routing mechanism enables automatic expert selection without relying on task identifiers and enhances interpretability.

**Weaknesses:**

1. Method: The method lacks both design discussion and experimental validation for handling the linear expansion of the expert pool as tasks increase.
2. Writing: The methodological description involves many equations and symbols, but several are insufficiently defined, making the approach difficult to follow and understand.

**Questions:**

1. Although the authors claim that computational cost is controlled via top-1 routing, the expert pool still grows linearly with the number of tasks.This may introduce additional computational overhead and lead to redundant learning or overfitting across experts.Will the proposed method remain effective as the expert pool expands?Is there a need for an explicit merging or consolidation mechanism to address this issue?
2. When anchors are updated, do the corresponding experts require retraining, and if so, how does this affect expert stability over time?
3.During anchor computation and expert selection, both visual and audio features are considered, but the paper does not explain how modality imbalance is handled. If one modality (e.g., visual) exhibits higher variance, could it dominate the routing decisions and introduce potential bias?

**Details Of Ethics Concerns:**

The paper acknowledges potential dataset bias and fairness risks in the Ethics Statement. No additional ethical concerns are observed.

---

### Official Review · Reviewer_9jBm · 2025-11-01

**Soundness:** 3
**Presentation:** 3
**Contribution:** 3
**Rating:** 6
**Confidence:** 4

**Summary:**

This paper introduces AVQACL-MoE to tackle catastrophic forgetting in Audio-Visual Question Answering Continual Learning (AVQACL). It works by freezing a strong multimodal backbone (CLIP, BEATs, LLaMA-2-7B). As new tasks arrive, the model adds and trains new, task-specific FFN expert modules, which are then frozen. This parameter isolation prevents forgetting. For inference, it avoids task labels by using "anchors"—a lightweight running-mean of each task's audio, visual, and text features. A parameter-free router picks the best expert by finding the highest cosine similarity between the input's features and the task anchors. The method posts state-of-the-art results on Split-AVQA and Split-MUSIC-AVQA, achieving high accuracy (86.41% / 72.02%) and almost no forgetting (2.34% / 1.79%) without data replay.

**Strengths:**

* Freezing the backbone and adding isolated, task-specific experts is a clean and effective way to stop catastrophic forgetting by design.

* The anchor-based router is a key strength. It's parameter-free, light on memory, and removes the need for task labels at inference, a common pain point.

* The model achieves state-of-the-art results, beating prior work by a large margin on both MA and AF.

* Getting these results with no data replay (M=0) is a big plus for efficiency and storage.

* The ablations are convincing. The "w/o CL" comparison proves the expert strategy is necessary, and the modality ablation clearly shows that tri-modal anchors are essential for routing.

**Weaknesses:**

* The modality ablation (Fig. 4) is revealing. When text is removed from the anchor, performance tanks (MA drops from 72% to 35% on one dataset). This implies the router is just identifying the question type from the text, not doing deep AV reasoning. The dataset splits (one task per question type) likely make this problem worse.

* Parameter count grows linearly with the number of tasks ($T$). Also, the routing cost isn't constant; it's $\mathcal{O}(T)$ because the input must be compared to all $T$ task anchors.

* Anchors are just a single running mean. This assumes tasks are simple and unimodal, which is unlikely to hold for complex tasks or tasks that drift over time.

* It's not clear if all baselines used the exact same powerful backbone (LLaMA-2-7B, CLIP-L, etc.). This makes it hard to know if the gains come from the new MoE method or just a stronger backbone.

**Questions:**

* What happens if you paraphrase the questions? If the router relies on keywords, performance will likely drop, showing it's not truly understanding the semantics.

* Why average the modality similarities equally? Did you try learning weights, so a question like "What instrument is playing?" could put more weight on audio?

* When does the $\mathcal{O}(T)$ routing cost become a real latency problem? (e.g., at T=25, 50, or 100 tasks).

* Why do you choose Qwen-Flash to score non-exact-match answers?

---

### Note · Authors · 2025-11-14

I have read and agree with the venue's withdrawal policy on behalf of myself and my co-authors.